# Study of a Mixed Conductive Layer Fabricated by Ion Implantation and Distribution Theory

**DOI:** 10.3390/polym15020270

**Published:** 2023-01-05

**Authors:** Xuerui Fan, Huiyan Zhang, Yi Wei, Yao Huang, Huimei He, Yun Wang, Qingyun Meng, Wenjie Wu

**Affiliations:** 1College of Mathematics and Physics, Beijing University of Chemical Technology, Beijing 100029, China; 2College of Material Science and Engineering, Beijing University of Chemical Technology, Beijing 100029, China; 3College of Mechanical and Electrical Engineering, Beijing University of Chemical Technology, Beijing 100029, China

**Keywords:** ion implantation, mixed conductive layer, distribution theory

## Abstract

Electrodes are essential parts of capacitors that can consist of a variety of materials depending on the application. In dielectric elastomer transducers (DETs)—a type of special variable capacitor—the electrode needs to deform with a soft base. However, the current carbon-based electrodes are not stable, and the metal-based ones are not flexible for use in DETs. Thus, the need to fabricate an electrode which can meet both the stability and flexibility requirements is extremely important. In this work, silver ions with energy levels of 40 keV were implanted into the surface of polydimethylsiloxane (PDMS) to explore the effect of ion implantation on surface conductivity. The experimental results showed that the surface resistivity of PDMS reached 251.85 kΩ per square and dropped by 10 orders of magnitude after ion implantation. This indicates that the surface conductivity was significantly improved. EDS characterization results showed that the maximum penetration depth that ions could reach was about 2.5 μm. The surface resistivity of the sample coated with carbon black was further reduced by an order of magnitude after ion implantation and changed more stably with time. A quasi-melting-collision model was established to investigate the distribution of carbon black particles. The concentration of carbon black particles at a distance from the PDMS surface followed a Gaussian-like distribution.

## 1. Introduction

Silicone rubber is widely used in all aspects of life [1,2,3,4,5] and is characterized by the advantages of high elasticity, non-flammability, and low cost. Polydimethylsiloxane (PDMS) is a typical silicone rubber material. Due to its low elastic modulus, high breakdown voltage, and high designability, PDMS is often applied in the fabrication of flexible electronic devices, such as dielectric elastomer generators (DEGs) [5,6,7,8,9] and dielectric elastomer actuators (DEAs) [10,11,12,13]. As lightweight and excellently flexible mechanical devices, DEAs and DEGs have drawn the attention of researchers in various fields, including those working on artificial muscles [13,14,15], soft robots [16,17], sensors [18], low-frequency generators [19], etc. The kernel modules of both DEAs and DEGs are dielectric elastomers sandwiched between two electrodes which alternate between large deformations and initial states [11,20,21]. Since they undergo considerable deformations during their operation for long periods of time, one of the main problems which remains to be solved concerning DEAs and DEGs is the fabrication of flexible electrodes with good adhesion and good conductivity after large amounts of deformation and which do not separate from the dielectric elastomers [12,22,23].

The electrodes commonly used for DEGs are fabricated using carbon nanotubes [24], graphene [25,26], carbon fiber [27], etc. [28]. Although they can be handily coated on dielectric elastomers to serve as carbon-based electrodes with high conductivities, these electrodes also have disadvantages. The large differences in moduli and the poor interfaces between carbon-based electrodes and DE materials lead to the poor durability of electrodes, on the one hand, and poor overall stability, on the other, which also constrain the deformation potentials of the materials. Thin metal film deposition can be used to fabricate patterned electrodes with high electrical conductivities [27,29,30]; however, this method also has disadvantages. First, the metal electrodes thus produced exhibit poor adhesion to elastomer surfaces, leading to delamination after several cycles of work deformation; secondly, fractures occur after deformations of a few percent are undergone. The Young’s modulus of a metal electrode is orders of magnitude higher than that of an elastomer, which significantly hinders the performance of flexible electronics.

Metal ions implanted into the surfaces of elastomers can form polymer–metal composite layers, causing nanometal particles to stay inside the elastomer, thus eliminating problems due to the phase interface between the electrode and the elastomer [31,32] and allowing the preparation of patterned electrodes by the addition of masks. Samuel Rosset [31] implanted low-energy ions obtained using filtered cathodic vacuum arc (FCVA) technology into PDMS and obtained a Au-PDMS mixed conductive layer with both low surface resistance and time stability. Nonetheless, this method has high requirements for the metal used, otherwise the time stability of the electrode will be relatively poor due to oxidation and degradation. Hence, the design and fabrication of an electrode with time and strain stability is of great research and practical value.

Herein, in order to obtain electrodes with high conductivities and stabilities, a metal vapor vacuum arc ion (MEVVA) source was employed to implant silver ions into a PDMS surface, which reduced the surface resistivity to 251.85 KΩ per square. In addition, this article proposes coating the surface of PDMS with a layer of carbon black before silver ion implantation, as this could serve as a method to tackle the problems related to conductive layer resistance time stabilities due to oxidation and degradation. A distribution model of carbon black particles in PDMS after ion implantation is also presented. More importantly, these methods can support the development of electrodes in other areas of dielectrics [33,34,35] and research [36,37,38]. This work has potential value for the design of new flexible electrodes and composite conductive functional materials.

## 2. Theoretical Model

Ion implantation affects the properties of polymer surfaces [39,40], and the degree of modification depends on implantation parameters, including implantation dose, accelerating voltage, ion beam intensity, etc. The surface composition and structure of a polymer undergo fundamental changes during the procedure which influence the depth distribution of implanted atoms, which in turn affects the structural, optical, electrical, and other properties of the polymer to certain degrees. Hence, it is of great significance to establish an appropriate theoretical model for the penetration depth distribution of implanted atoms before performing implantation experiments. The theoretical model of implantation proposed by Lindhard [41] in 1963 concerned the interactions between implanted ions and the target directly and explained the penetration distance and distribution of implanted ions accordingly. Subsequent theoretical expansions continued to focus on the penetration distance and distribution of implanted ions during mask implantation, yet there was no further discussion of the penetration depth of coated materials on the targets. For this reason, in this report, a quasi-melting-collision model is proposed to try and solve the problem of the distribution of carbon black particles under the surface of a polymer.

### 2.1. Initial Model

The experiments in this work used silver plasma produced from the MEVVA source to form an ion beam after passing through an accelerated electric field that interacts with PDMS in the target chamber.

In this model, it is assumed that the direction of silver ion movement is completely perpendicular to the plane direction of the target and that the beam is uniform. The energy obtained by the silver ions after passing through the accelerating electric field is fully converted into the kinetic energy of the silver ions, and the velocity of the silver ions, according to $v_{ag}=\sqrt{\frac{2qU}{m_{1}}}$, is:(1)$$v_{ag}=\sqrt{\frac{2qU}{m_{1}}}$$
where $v_{ag}$ is the silver ion average velocity, q is the silver ion charge quantity, U is the accelerating voltage, and $m_{1}$ is the mass of the silver ions.

TEM images of carbon black particles showed that they formed spherical aggregations at the microscopic level. In order to simplify the calculation process, we assumed that the carbon black particles were rigid balls and that their internal structures did not change when the ion beam was applied to them.

We also assumed that the silver ions in the ion beam were free and that the carbon black particles were kept in the quiescent state on the surface of the PDMS. For the purpose of exploring the penetration depth of the carbon black particles, the collision between the silver ions and the carbon black particles was treated as the central collision, and it was assumed that the collision would follow the law of conservation of momentum and energy. So, the maximum velocity of the carbon black particles after the collision is:(2)$$v_{cmax}=\frac{2m_{1}}{m_{1}+m_{2}}v_{1}$$
where $v_{cmax}$ is the velocity of carbon black particles after the collision, $m_{2}$ is the mass of carbon black particles, and $\;v_{c}=v_{cmax}$.

The motion process of carbon black particles and silver ions under the polymer surface can be regarded as the motion of spherical objects overcoming the resistance in viscous laminar flow. According to Stokes’ law, [30] the resistance of silver ions in the polymer at a certain time is:(3)$$f_{1}=6\pi \eta r_{1}v_{1}$$
where $f_{1}$ is the viscous resistance of the silver ions, η is the surface viscosity coefficient measured on the PDMS surface by dynamic mechanical analysis (DMA), $r_{1}$ is the radius of the silver ions, and $v_{1}$ is the velocity of the silver ions at the time.

According to $k_{1}=6\pi \eta r_{1}$ Newton’s second law, the velocity changes for individual silver ions entering the polymer surface follows:(4)$$6\pi \eta r_{1}v_{1}=-m_{1}\frac{dv_{1}}{dt}$$

Due to $\frac{dv_{1}}{dt}=\frac{dv_{1}}{dx}\frac{dx}{dt}=\frac{dv_{1}}{dx}v_{1}$, Equation (4) can be transformed into:(5)$$6\pi \eta r_{1}dx=-mdv_{1}$$

Integrating both sides of Equation (4), the velocity of a single silver ion after entering the polymer changes with time:(6)$$v_{1}\left(t\right)=v_{ag}e^{-\frac{6\pi \eta r_{1}}{m_{1}}t}$$

When the silver ions stay in the polymer, the distances that the silver ions travel are the penetration depths under the surface of the polymer. Integrating time on both sides of Equation (6), the total penetration depth of carbon black particles after collision with a single silver ion, $s_{1}$, is:(7)$$s_{1}=\int_{0}^{+\infty }v_{1}\left(t\right)dt$$

Equation (8) can be obtained by integrating Equation (5) on both sides.
(8)$$v_{1}\left(t\right)=v_{ag}-\frac{k_{1}}{m_{1}}x_{1}$$

Equation (8) is helpful in calculating the moving speed of silver ions at the time according to the moving distance of silver ions in the polymer in the subsequent model.

We monitored the change in the resistivities of silver ions directly implanted into the PDMS over time, as shown in next section. It could be seen that the surface conductivity of PDMS improved saliently after implantation, but the conductivity was not stable over time, and the longer the time after implantation, the worse the conductivity. Carbon black particles are relatively stable conductive substances. For this reason, it was proposed to coat a layer of carbon black on the surface of the PDMS before ion implantation.

Now, let us calculate the motion of carbon black particles. The resistance of carbon black particles is:(9)$$f_{2}=6\pi \eta r_{2}v_{2}$$
where $f_{2}$ is the viscous resistance of carbon black, $r_{2}$ is the carbon black particle radius, $v_{2}$ is the velocity of carbon black particles at the time, and $k_{2}=6\pi \eta r_{2}$.

According to Newton’s second law, the velocity changes of individual carbon black particles entering the polymer surface follow:(10)$$6\pi \eta r_{2}v_{2}=-m_{2}\frac{dv_{2}}{dt}$$

Hence, the velocities of carbon black particles change with time:(11)$$v_{2}\left(t\right)=v_{c}e^{-\frac{6\pi \eta r_{2}}{m_{2}}t}$$

Integrating Equation (11) with time, the distance that a single carbon black particle moves after colliding with a single silver ion is:(12)$$s_{2}=\int_{0}^{+\infty }v_{2}\left(t\right)dt$$

The relation between the forward distance of carbon black particles and velocity at the time is:(13)$$x_{2}=\int_{0}^{t}v_{2}e^{-\frac{6\pi \eta r_{2}}{m_{2}}}dt=\frac{m_{2}}{k_{2}}v_{c}-\frac{m_{2}}{k_{2}}v_{2}\left(t\right)$$

Equation (13) is helpful in calculating the movement distance of a single carbon black particle when it collides with successive silver ions in subsequent models.

Due to the continuous collisions of individual carbon black particles with silver ions in the ion beam, it is necessary to determine the number of silver ions that collide with a single carbon black particle and their motion states at the time of collision when the implantation dose is fixed under given experimental conditions if we are to calculate the total penetration depth of a single carbon black particle.

To start with, let us discuss the maximum number of silver ions that can collide with a single carbon black particle. The implantation dose N refers to the number of implanted ions received per unit area, $s_{0}$, in ion implantation. Under the condition of a uniform beam current, the maximum number of silver ions received by a single carbon black particle in the area occupied by the polymer plane projection is $n_{0}=\frac{\pi r_{2}^{2}}{s_{0}}N$.

Next, let us get down to the state problem of a single carbon black particle in collision with continuous silver ions. First of all, the time difference $t_{0}$ between the continuous silver ions in the ion beam reaching the polymer surface must be calculated. The beam current density, J, refers to the charge in unit area passing through the point in the direction perpendicular to the positive charge in unit time. In the ion implantation process, the unit of J is $\mu A/{\mathrm{cm}}^{2}$, while the unit area $s_{0}=1\times 1\;{\mathrm{cm}}^{2}$. So, $I=Js_{0}$ is the current intensity per unit area.

In a cuboid with area base $\;s_{0}$ and height $v_{ag}dt$, the magnitude of current I is:(14)$$I=\frac{s_{0}v_{ag}qn_{1}dt}{dt}$$
where $n_{1}$ is the actual number of silver ions per unit volume in the ion beam, $n_{1}=\frac{I}{v_{ag}qs_{0}}$.

The beam intensity of the ion beam $J^{\prime }$ is the number of charges passing through a point per unit time and projected area, $J^{\prime }=n_{1}v_{ag}$.

The number of silver ions received in the area of a single carbon black particle projected onto the polymer plane in unit time is $n_{2}=\frac{\pi r_{2}^{2}}{s_{0}}J^{\prime }$; hence, $t_{0}=\frac{1}{n_{2}}=\frac{s_{0}}{\pi r_{2}^{2}J^{\prime }}$.

After a single carbon black particle collides with the first silver ion in the ion beam reaching the target surface, it moves through the polymer in the direction perpendicular to the surface of the polymer. Before the second silver ion reaches the polymer surface, the carbon black particle has been moving for a time $t_{0}$, and the forward distance is $x_{c1}$. The speed of silver ions is several orders of magnitude higher than that of the carbon black particle. It can be calculated, using Matlab R2019a, that the time before the silver ion collides with the carbon black in the polymer reaches $10^{-16}$ s; therefore, the time of its movement in the polymer is ignored here. When the carbon black particle moves for $t_{0}$, the second silver ion enters the polymer and collides with the carbon black particle elastically.

At this time, the distance, $x_{c1}$, and the velocity, $v_{c12}$, of the carbon black particle before the collision with the second silver ion are, respectively:(15)$$x_{c1}=\frac{m_{2}}{k_{2}}v_{c11}\left(1-e^{-\frac{k_{2}}{m_{2}}t_{0}}\right)\left(v_{c11}=v_{c}\right)$$
(16)$$v_{c12}=v_{c11}e^{-\frac{k_{2}}{m_{2}}t_{0}}$$

The distance and the velocity of the second silver ion in the polymer are, respectively:(17)$$x_{ag2}=x_{c1}$$
(18)$$v_{ag2}=v_{1}-\frac{k_{1}}{m_{1}}x_{ag2}$$

The second silver ion might collide with the carbon black particle if $v_{c12}<v_{ag2}$. Assuming that the silver ions colliding with the carbon black particle are free, according to the law of energy and momentum conservation, the velocity of the carbon black particle after colliding with the second silver ion is:(19)$$v_{c21}=\frac{2m_{1}v_{ag2}+\left(m_{2}-m_{1}\right)v_{c1}}{m_{1}+m_{2}}$$

Since the time for the movement of silver ions in the polymer is ignored, the carbon black particle continues to move in the polymer after colliding with the second silver ion in the direction perpendicular to the horizontal plane of the polymer and might collide with the third silver ion. The forward distance of the carbon black particle after the collision with the second silver ion and the velocity before the collision with the third silver ion are, respectively:(20)$$x_{c2}=\frac{m_{2}}{k_{2}}v_{c21}\left(1-e^{-\frac{k_{2}}{m_{2}}t_{0}}\right)$$
(21)$$v_{c22}=v_{c21}\left(1-e^{-\frac{k_{2}}{m_{2}}t_{0}}\right)$$

The distance traveled by the third silver ion and the velocity before the collision with the carbon black particle are, respectively:(22)$$x_{ag3}=x_{c1}+x_{c2}$$
(23)$$v_{ag3}=v_{1}-\frac{k_{1}}{m_{1}}x_{ag3}$$

The third silver ion collides with the carbon black particle if $v_{c22}<v_{ag3}$. According to the law of conservation of energy and momentum, the velocity of the carbon black particle and the third silver ion after collision is:(24)$$v_{c31}=\frac{2m_{1}v_{ag3}+\left(m_{2}-m_{1}\right)v_{c22}}{m_{1}+m_{2}}$$

According to mathematical derivation, before colliding with the n+1th silver ion, the carbon black particle moves in the polymer for $t_{0}$ in the direction perpendicular to the horizontal plane of the polymer. It is assumed that the n+1th silver ion can collide with the carbon black particle. The forward distance of the carbon black particle after colliding with the nth silver ion and the velocity before the collision with the n+1th silver ion are respectively:(25)$$x_{cn}=\frac{m_{2}}{k_{2}}\left(1-e^{-\frac{k_{2}}{m_{2}}t_{0}}\right)v_{cn1}$$
(26)$$v_{cn2}=v_{cn1}\left(1-e^{-\frac{k_{2}}{m_{2}}t_{0}}\right)$$

The distance and the velocity of the n+1th silver ion are, respectively:(27)$$x_{ag\left(n+1\right)}=x_{c1}+x_{c2}+\cdots x_{cn}$$
(28)$$v_{ag\left(n+1\right)}=v_{1}-\frac{k_{1}}{m_{1}}x_{ag\left(n+1\right)}$$

In order to determine whether the n+1th silver ion can collide with the carbon black particle in this time, the size of $v_{ag\left(n+1\right)},v_{cn2}$ needs to be compared. If $v_{ag\left(n+1\right)}\gg v_{cn2}$, the situation of the collision between the carbon black particle and the next silver ion can be calculated. If $v_{ag\left(n+1\right)}<v_{cn2}$, the carbon black particle will not collide with the n+1th or subsequent silver ions, and since the implantation dose, N, was fixed during the experiment, n should obey $n\leq n_{0}$.

The total penetration depth of a single carbon black particle is:(29)$$X=\overset{n-1}{\underset{i=1}{\sum }}x_{ci}+\int_{0}^{+\infty }v_{cn1}dt$$

Evidently, the total penetration depth, X, of a single carbon black particle is a function of U, N, and r. When any two variables are fixed, the change relationship of the other quantity can be obtained. We used MATLAB R2019a to simulate the relationship between the total penetration depth of a single carbon black particle and U, N, and r.

Figure 1 showed the change in total penetration depth, X, of a single carbon black particle with accelerating voltage U obtained by Matlab simulation. In Figure 1a, the size of r was fixed at 10 nm, and the implantation dose was $1\times 10^{13}-1\times 10^{15}$ ions/cm^2^. It can be seen distinctly from Figure 1a that X increased with U and that the implantation dose did not affect this trend. However, with the increase in voltage, the change rate of total penetration depth X decreased, and the contribution of the same voltage increase to the increase in penetration depth decreased, indicating that it is not necessary to pursue an excessive accelerating voltage when selecting experimental conditions. In Figure 1b, the implantation dose was fixed at $1\times 10^{17}$ ions/cm^2^, r was from 10 nm to 50 nm, and the five ***X***-***U*** curves were intuitively overlapped, indicating that under such experimental conditions the size of r had little influence on the total penetration depth. Similarly, the same conclusion, in accordance with the left picture, that the larger the U, the larger the X, can be derived.

Figure 2 shows variation in the X of a single carbon black particle with the N obtained by MATLAB R2019a simulation. In Figure 2a, r is fixed at 10 nm, and the accelerating voltage goes from 10 kV to 50 kV. As N increases, the total penetration depth increases first, then tends to take a constant value when N reaches a threshold. The accelerating voltage during implantation only affects the constant value and has no effect on the change trend, which is consistent with the trend shown in Figure 2a. As the implantation dose increases in Figure 2a, the ***X***-***U*** curve shows signs of superposition, which also indicates that increasing the implantation dose has no effect on the total penetration depth when the implantation dose reaches a certain threshold. In Figure 2b, the accelerating voltage is fixed at 40 kV, and r varies from 10 nm to 50 nm. The conclusion is consistent with Figure 2a.

We investigated the effect of carbon black particle size on the total penetration depth, X, of a single carbon black particle. In Figure 3a, the accelerating voltage was 40 kV, and the implantation dose was $1\times 10^{13}-1\times 10^{15}$ ions/cm^2^. When the implantation dose was $1\times 10^{13}-1\times 10^{14}$ ions/cm^2^, the total penetration depth, X, was tremendously affected by r, increasing with r and then finally tending to become stable. When the dose was $1\times 10^{14}-1\times 10^{15}$ ions/cm^2^, the total penetration depth, X, remained constant, indicating that r hardly had any effect on it. Therefore, it can be concluded that r has nearly no effect on the total penetration depth, X, when N is large enough. In Figure 3b, the implantation dose was constant at $1\times 10^{17}$ ions/cm^2^, and the accelerating voltage varied from 10 kV to 50 kV. The ***X***-***r*** curve is approximately a straight line parallel to the X-axis, which is consistent with the conclusion obtained with respect to the left figure, that is to say, r has no influence on X under such conditions.

To sum up, in the model proposed in Equations (1) to (29), carbon black particles finally gather at the farthest location from the polymer surface under an ideal set of conditions. For this reason, the model needs to be modified and improved.

### 2.2. Model Modification

In order to explore the specific distribution of carbon black particles after entering the surface, we need to modify the model described in Section 2.1.

#### 2.2.1. Amendment of *N*

In the model establishment and the MATLAB R2019a simulation of Section 2.1, it was assumed that the ion beam is uniform, but the beam density cannot be uniform in the actual implantation process. According to measurement results reported in the literature [42,43], it can be known that the distribution follows Gauss’s law, which means that the density in the middle part is the highest and that the density decreases towards the two side parts. The target of ion implantation was a disk with a diameter of 16 cm, so a probe into the influence of the inhomogeneity of the beam density was needed. We quoted the measurement results of Zhu [44] and then used MATLAB for fitting to obtain the beam density distribution function with an arbitrary radial distance, as shown in Figure 4a.

In Figure 4b, r was fixed at 10 nm and the accelerating voltage was 40 kV to show the total penetration depth distribution of carbon black particles in the radial direction under different implantation doses when the beam density followed the distribution conditions shown in Figure 4a. With a fixed implantation dose, the distribution of the penetration distance of carbon black particles near the center of the circle was uniform, and the farther to the edge, the lower the penetration distance. The region of uniform distribution gradually increased with the increase in implantation dose, but the rate at which the region increased decreased accordingly. In order to ensure the uniformity of carbon black distribution after implantation, the implantation dose was chosen to be $1\times 10^{17}$ ions/cm^2^, and the implantation area was set as a square with a side length of 10 cm.

#### 2.2.2. Amendment of *U*

In the model establishment and MATLAB simulation of Section 2.1, it was assumed that the silver ions had only one speed after the accelerating voltage, meaning there were only monovalent silver ions, whereas, as a matter of fact, sliver ions with other valence states also existed in the ion beam extracted from MEVVA. For an accelerating voltage of 40 kv, the valence states and the proportions of each valence state are shown in Table 1.

Hence, corrections need to be made to the initial velocity of silver ions in the calculation model. From both Figure 2 and Figure 4, the conclusion that there is no effect of the implantation dose on the total penetration depth of individual carbon black particles when the dose exceeds the threshold can be drawn. In the collision process, N affects the theoretical size of the total number of collisions with a single carbon black particle. We demonstrated the relationship between the total penetration depth of a single carbon black particle and the total number of collisions when the accelerating voltage was 40 kV, and the radius was 5 nm. The corresponding results are shown in Figure 5.

Figure 5 shows that the total penetration depth of a single carbon black particle increased initially and then tended to become constant as the total number of collisions increased. Silver ions with different energies showed the same change trend, meaning that n in Equation (29) can be taken as 150 under such conditions. Equation (29) can be rewritten as:(30)$$X=\overset{150}{\underset{i=1}{\sum }}x_{ci}+\int_{0}^{\infty }v_{c1501}dt$$

The energy levels of silver ions before each collision with carbon black particles in Equation (30) are different from the levels in Equation (29), which were, randomly, either 40 keV, 80 keV, 120 keV, or 160 keV. Therefore, we could first calculate the occurrent number of silver ions with different energies among the 150 silver ions and the probability at the time and then calculate the maximum total penetration distance under different conditions. We ignored the silver ions with 160 keV in our calculation because their numbers were too low. Since the probability of the initial energy of a single silver ion was fixed, we could use a multinomial distribution to calculate this event. The initial energy of each silver ion colliding with a carbon black particle might have three possible outcomes, $A_{1}$, $A_{2}$, or $A_{3}$. ($A_{1}$ represents 40 keV, $A_{2}$ represents 80 keV, and $A_{3}$ represents 120 keV). Their probabilities are $P_{1}$, $P_{2}$, and $P_{3}$ ($P_{1}=0.13$, $P_{2}=0.65$, $P_{3}=0.22$), respectively. In the results for the total sampling of 150 times, the probability of occurrence of $n_{1}$ times for $A_{1}$, $n_{2}$ times for $A_{2}$, and $n_{3}$ times for $A_{3}$ is:(31)$$P(A_{1}=x_{1},A_{2}=x_{2},A_{3}=x_{3})=\frac{150!}{x_{1}!x_{2}!x_{3}!}p_{1}{}^{x_{1}}p_{2}{}^{x_{2}}p_{3}{}^{x3},\;\mathrm{when}\;n_{1}+n_{2}+n_{3}=150,n_{i}\geq 0$$

There are 11,476 events in total, as shown in the first four columns of Table 2.

We used MATLAB to calculate the maximum total penetration depths of a single carbon black particle for all the events in Table 2. The fifth and sixth columns of Table 2 show the penetration distances and the probabilities of the corresponding events, respectively.

We arranged the data in fifth and sixth columns in Table 2 according to the data in the fifth column from smallest to largest, as represented by the black curve in Figure 6. The horizontal axis represents the total penetration depth of a single carbon black particle, while the vertical axis represents the corresponding probability. Figure 6 shows the probability distribution for the penetration depth of a single carbon black particle, and the red curve shows the result after fitting. It can be found that when the radius was 5 nm, the probability distribution of the penetration depth of a single carbon black particle followed a Gaussian-like distribution.

When the implantation dose was $1\times 10^{17}$ ions/cm^2^, the number of silver ions theoretically received by the polymer on the projected area of a single carbon black particle was considerably more than 150. Therefore, the radius has no effect on the farthest distance that the carbon black particles can reach, and the selected concentration has no effect on the total penetration depth of a single carbon black particle. So, the probability distribution for the penetration depth of a single carbon black particle shown in Figure 6 is also the distribution for the concentration of carbon black particles in the polymer phase (the number of particles in the unit volume) with the penetration distance.

## 3. Experimental Section

### 3.1. Experimental Materials

The PDMS used in this paper was purchased from a Chinese science experimental materials store. The carbon black for the coating, Ketjen Black High Conductive Carbon Black EC-600JD, was produced by the Lion King Corporation of Japan (Tokyo, Japan). The properties of the materials are shown in Table 3, but not all.

### 3.2. Preparation of Samples and Experimentation

Samples with a size of $5\times 10{\;\mathrm{cm}}^{2}$ were prepared and then cleaned ultrasonically with deionized water, ethanol, and acetone in turn for 15 min before the experiment. Then, they were taken out and put into a drying box to be dried at 50 °C for an hour to remove the surface adsorbents, after which they were cooled to room temperature and put into the ion implanter cavity for implantation experiments.

Combined with the calculations of the theoretical model and the actual experimental conditions, we chose the silver ion with the best conductivity as the implanted ion. The implantation dose was $1\times 10^{17}$ ions/cm^2^, the energy was 40 keV, the beam current was 0.8 $\mathrm{mA}/{\mathrm{cm}}^{2}$, and the vacuum was $4\times 10^{-3}$ Pa.

### 3.3. Testing Instruments

The surface resistance of the sample was measured with an RTS-8 four-probe tester. The surface morphologies of the samples were analyzed using a HITACHI S-4800 scanning electron microscope (Hitachi Company, Tokyo, Japan), with a working voltage of 5 V. A Hitachis-4700 was used for the X-ray energy spectrum analysis of the sample, with a working voltage of 20 kV. The infrared absorption spectra of the samples were analyzed using the Tensor 27 infrared spectrometer produced by Bruker, Germany, with a scanning range of 4000–600 cm^−1^, to analyze the structural changes in the samples under different treatment processes.

## 4. Results and Discussion

### 4.1. Morphologies and Electrical Performances of the Samples after Silver Ion Implantation

#### 4.1.1. Changes in the Morphologies and Conductivities of the Sample Surfaces after Silver Ion Implantation

Samples 1, 3, and 5 consisted of PDMS implanted directly with silver ions with energies of 30 keV, 40 keV, and 50 keV, and samples 2, 4, and 6 consisted of PDMS coated with carbon black initially followed by implantation with silver ions with energies of 30 keV, 40 keV, and 50 keV. After implantation, there was a distinct silver film on the surface of sample 1. The surface conductivity of sample 1 was 251.85 kΩ per square, which was 10 orders of magnitude lower than the original sample, indicating that there was a conductive layer on the surface. To verify the stability of the conductive layer, the resistivity of the fixed point of the sample was measured at a fixed time, daily, for 15 days. It can be seen from Figure 7 that the resistivity increased with time. The samples with implantation energies of 40 keV and 50 keV showed the same pattern.

After 15 days, the surface resistance of the sample was nearly twice that of the original samples. This was because silver ions and PDMS form a blended structure on the surface of the latter after implantation, and, due to the oxidization of silver ions, the surface carrier concentration decreased while the resistivity increased with time. For this reason, it is of great significance to further introduce substances that are not easily oxidized to make the conductivity more stable in order to tackle the problem of the time instability of conductivity.

As shown in Figure 8, under a magnification of 10 K, the overall surface of the sample was flat and uniform, and there were small particles distributed on the surface. At a magnification of 200 K, it can be seen that there were tiny gaps on the surface. This was due to the heating of the polymer surface caused by ion implantation, which leads to the melting of the microdomain. However, due to thermal expansion and contraction, the molten state of the surface appeared after returning to room temperature. In Figure 7, the surface resistivity of sample 2 was 120.65 kΩ per square, which was one order of magnitude lower than that of the direct implantation, meaning that the conductivity was significantly improved, therefore indicating that the carbon black particles had indeed entered the surface of the sample. Moreover, the surface resistivity of sample 2 changed slowly with time, meaning that the conductivity exhibited a good time stability. It can be seen from Figure 8b that the block structure area of sample 2 was larger than that of Figure 8a. Samples 4 and 6 also showed the same pattern. This was because some of the silver ions collided with the carbon black particles first and the damage was relatively small compared with direct contact with the surface of the sample.

#### 4.1.2. Dielectric Property Analysis

In addition to the stability of the surface conductivity, the dielectric properties of the film after implantation are also very important. From the general calculation formula for power generation, the higher the dielectric constant, DE, of a material is, the better the power generation performance, theoretically. Therefore, the dielectric constant is measured in this section, as shown in Figure 9.

It can be observed in Figure 9a that the dielectric constant of the original PDMS increased with increasing frequency at low frequencies and that it tended to be stable after 200 Hz, at about 3.1. When the acceleration voltage was 30 kV, the dielectric constants of the PDMS samples after silver ion implantation increased to a certain degree and also tended to be stable after 200 Hz, reaching 3.15. However, the dielectric constant of the samples coated with carbon black re-injection increased at low frequencies, reaching 3.85 at 40 kV, and then gradually decreased with the increase in the test frequency and finally tended to become stable. The dielectric constant of sample 3 in Figure 9b is higher than that of sample 1 in Figure 9a after stabilizing, reaching 3.25. This may be due to the increase in the acceleration voltage during ion implantation, which leads to a wider distribution of silver ions in PDMS films and thus a better promotion effect. Figure 9c,d show the dielectric loss of the sample after injection. It can be seen that the dielectric loss is high at a low frequency, but after stabilizing at a high frequency the dielectric loss is less than 10^−1^, indicating that the conductivity loss is also low. In summary, ion implantation can improve the dielectric properties of silicone rubber slightly, and the dielectric loss meets the requirements of power generation.

### 4.2. Structural Analysis and Material-Phase Analysis

#### 4.2.1. Infrared Structure Analysis

It can be seen from Figure 10 that, after ion implantation, the PDMS surface had the original characteristic peaks, indicating that the polymer surface after implantation had a mixed structure composed of the raw materials used—silver ions and carbon black particles.

From the comparison of the three samples in the figure, it can be seen that the position of the Si-C stretching vibration peak of wave number 793, the asymmetric stretching vibration peaks of 1015 and 1078 of Si-O-Si, and the peak of CH_3_ of 1259 did not change significantly, but the absorption intensity decreased to a certain extent, meaning that during the process of ion implantation, these three bonds and groups were damaged to some degree. The Si-O-Si bond on the main chain did not suffer too much damage; mainly those on the branched chain did. The peaks for the samples coated with carbon black decreased to lesser extents, which indicated that carbon black played a certain role in shielding. From the above results, it can be seen that both silver ions and carbon black particles had entered the surface of the PDMS. The distribution of carbon black particles under the surface of the PDMS is discussed in Section 4.2.2.

#### 4.2.2. Energy Dispersive Spectroscopy Analysis

It can be seen from Figure 11a that the concentration of silver ions at a depth of 5 microns first increased and then decreased. Figure 11b shows the energy spectrum of a cross section of sample 4, in which the change trend for silver ions was the same as that in Figure 11a within 2.5 microns. The distance became smaller due to the hindering effect of carbon black particles, resulting in a decrease in the depth at which silver ions entered and a faster downward trend than in the left figure.

Evidently, the change in carbon concentration was due to the coating on the sample surface being bombarded and ions entering and staying under the surface of the PDMS. Figure 11a shows the linear relative concentration change in carbon elements in the cross sections of samples 3 and 4. The linear carbon concentration of sample 3 increased initially and then decreased from a depth of 0 μm to 2 μm due to the densification of carbon caused by ion implantation, which was consistent with previous reports in the literature [45]. The carbon concentration in sample 4 also showed the same trend from 0 to 5 microns, with a peak at 2.5 microns, which was due to the change in carbon concentration caused by carbon black particles staying in the polymer during ion implantation.

The concentration curve for carbon black particles with distance was obtained and presented in Figure 12a using peak separation. It also showed an increasing concentration distribution at first, which then decreased from 2 to 6 microns, similar to that in Figure 12b. The distribution trend for the calculated carbon black particle concentration was consistent, proving that the change in carbon concentration was caused by the carbon black particles. The EDS results verified the derivation in Section 2.2.2, which might provide theoretical guidance for the design of conductive materials. The experimental data and the calculated results are consistent with respect to the change trend, but the half-width (FWHM) of the carbon black particle concentration in the experiment was larger than that of the calculated results. This is because the collision between carbon black particles and silver ions was simplified as a positive collision in the established quasi-melting-collision model, but in practical terms the collision between silver ions and carbon black particles is very complicated, which will also affect the polymerization of carbon black particles.

## 5. Conclusions

In summary, the quasi-melting-collision model established for ion implantation of PDMS coated with carbon black on the surface was used to calculate the concentration distributions of carbon black particles in cross sections of PDMS, which showed Gaussian-like distributions. The parameters influencing concentration distribution include accelerating voltage, implantation dose, and particle energy distribution in the ion beam. The surface conductivity of PDMS after implantation was improved, and the material coated with carbon black had better conductivity and stability than the uncoated material, which might enhance the application value of the ion implantation method. The FTIR and EDS results showed that the implanted ions and the carbon black coated on the surface of the PDMS could cause the elemental contents on the PDMS surface to change at a depth of about 5 μm, and the vibration of the branched chain was mainly affected by the ion implantation. Carbon black particles and silver ions were embedded in the surface of the polymer, and the concentration distribution followed a Gaussian distribution, which was consistent with the calculated results. Therefore, the theory of ion implantation and a method for constructing a highly durable conductive layer on the surface of insulating materials has been developed in this paper. Such findings might have potential applications in the fields of sensors, dielectric elastomer generators, and conductive polymer materials.

## Figures and Tables

**Figure 1 polymers-15-00270-f001:**
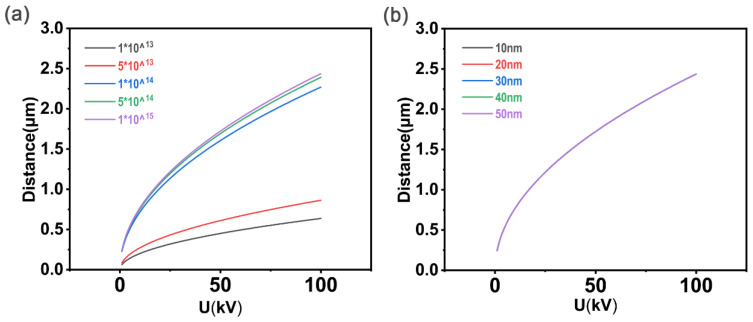
(**a**) ***r*** = 10 nm: the total penetration depth of the individual carbon black particle varies with the accelerating voltage. (**b**) ***N*** = $1\times 10^{17}$ ions/cm^2^: the total penetration depth of the individual carbon black particle varies with the accelerating voltage.

**Figure 2 polymers-15-00270-f002:**
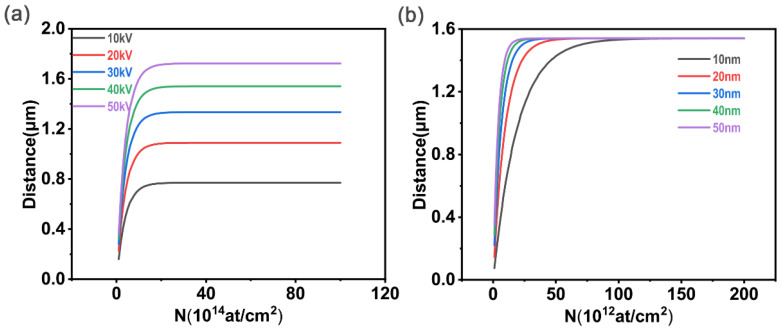
(**a**) ***r*** = 10 nm: the total penetration depth of the individual carbon black particle varies with the implantation dose. (**b**) ***U*** = 40 kV: the total penetration depth of the individual carbon black particle varies with the implantation dose.

**Figure 3 polymers-15-00270-f003:**
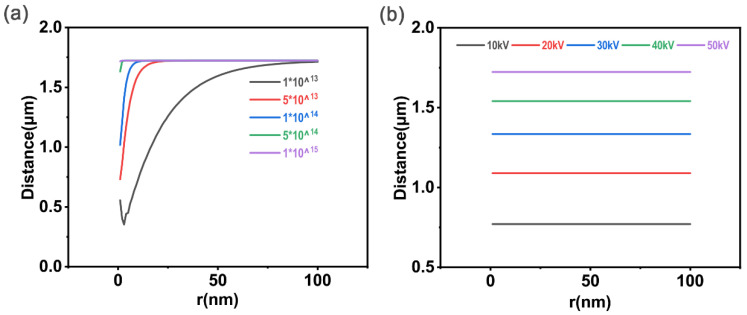
(**a**) ***U*** = 40 kV: the total penetration depth of the individual carbon black particle varies with carbon black particle size. (**b**) ***N*** = $1\times 10^{17}$ ions/cm^2^: the total penetration depth of the individual carbon black particle varies with carbon black particle size.

**Figure 4 polymers-15-00270-f004:**
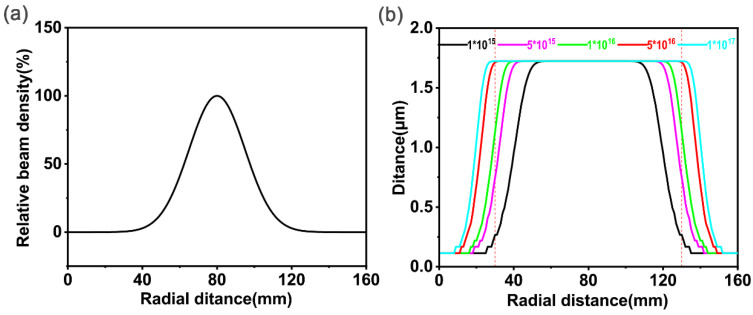
(**a**) Ion beam density distribution. (**b**) Ion beam density distribution’s influence on the total penetration depth of carbon black particles.

**Figure 5 polymers-15-00270-f005:**
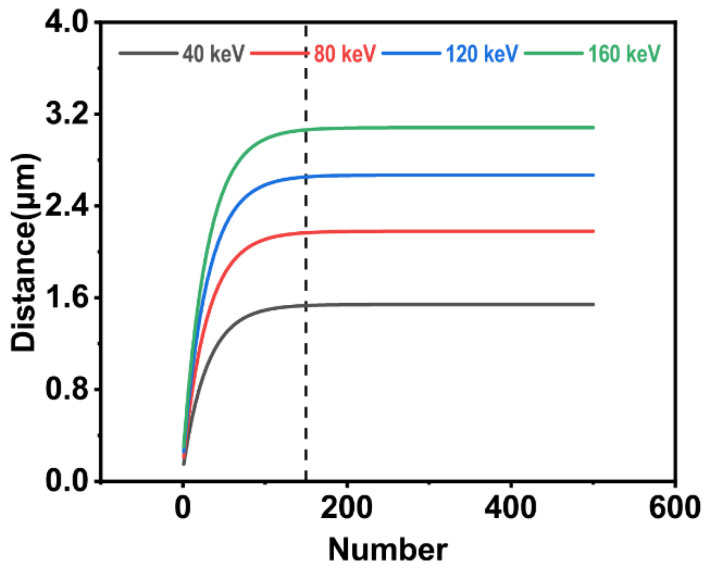
Relationship between the total penetration depth of a single carbon black particle and the total number of collisions.

**Figure 6 polymers-15-00270-f006:**
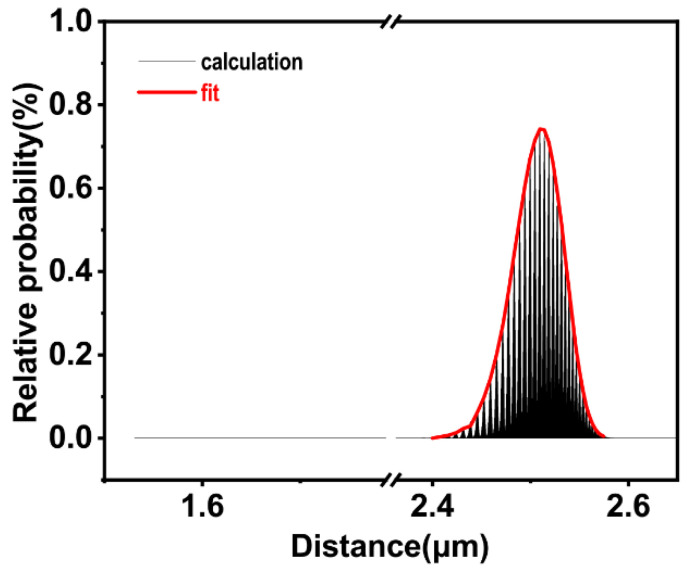
Probability distribution of the total penetration distances of the individual carbon black particles.

**Figure 7 polymers-15-00270-f007:**
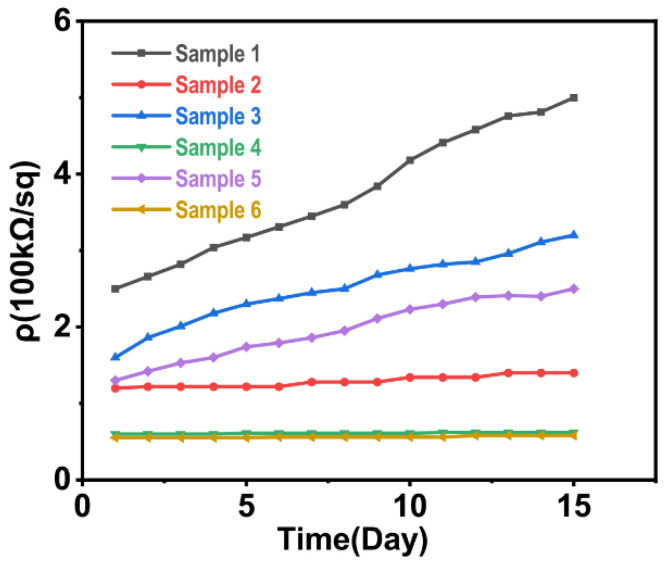
Changes in the surface resistivities of samples 1–6 with time.

**Figure 8 polymers-15-00270-f008:**
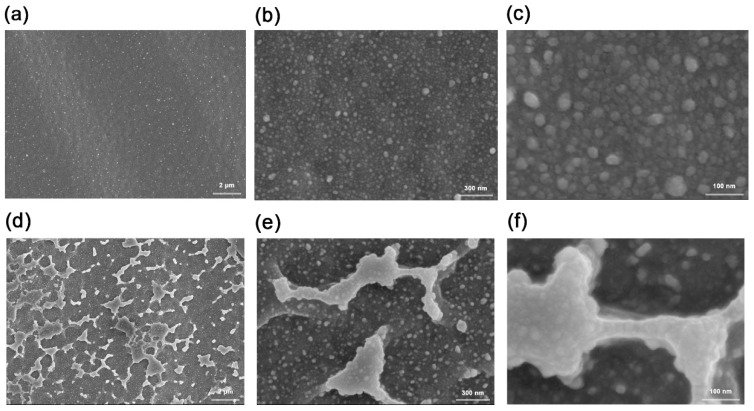
(**a**–**c**) The 10 K, 60 K, and 200 K SEM photographs of sample 3. (**d**–**f**) The 10 K, 60 K, and 200 K SEM photographs of sample 1 and sample 4.

**Figure 9 polymers-15-00270-f009:**
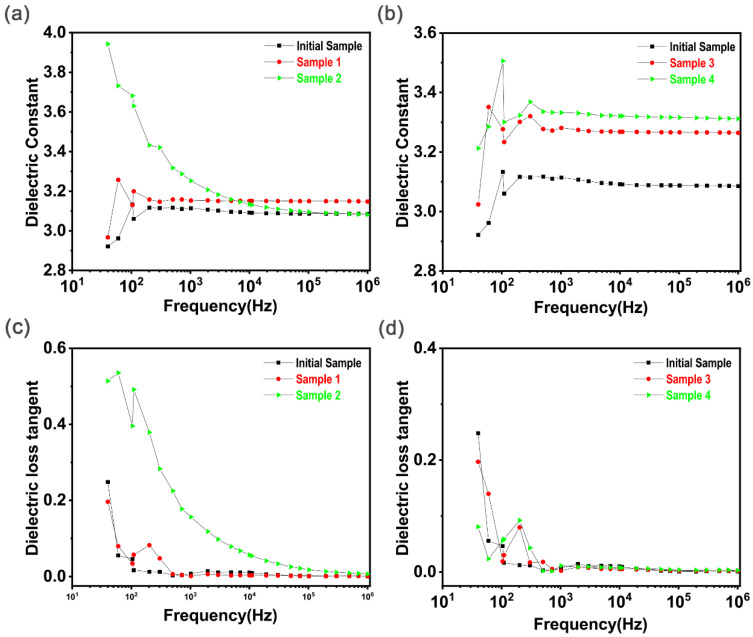
(**a**,**c**) The dielectric constant and dielectric loss test for the original sample and samples 1 and 2 at different frequencies. (**b**,**d**) The dielectric constant and dielectric loss test for the original sample and samples 3 and 4 at different frequencies.

**Figure 10 polymers-15-00270-f010:**
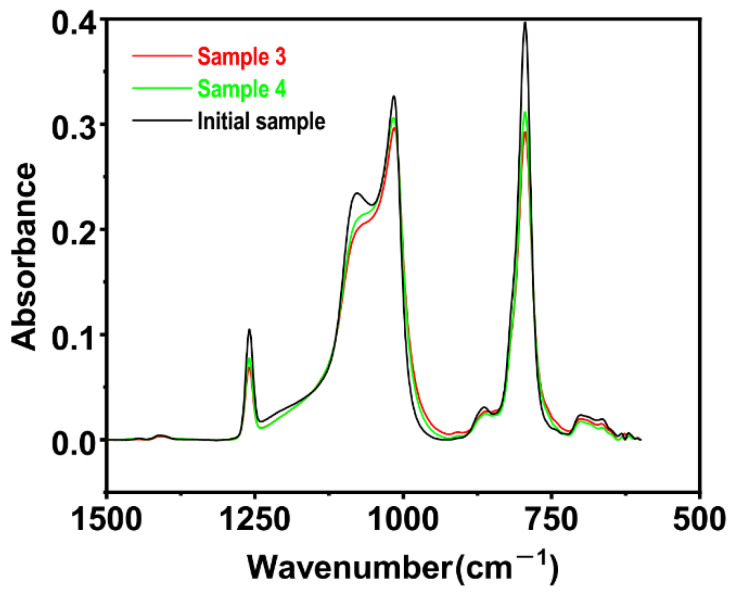
Infrared absorption spectra for the initial sample and samples 3 and 4.

**Figure 11 polymers-15-00270-f011:**
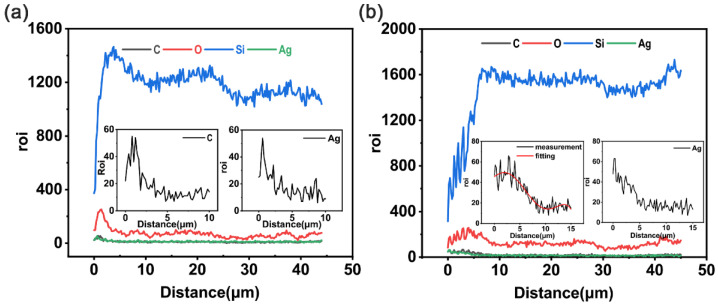
(**a**) EDS image of the cross section of sample 3. (**b**) EDS image of the cross section of sample 4.

**Figure 12 polymers-15-00270-f012:**
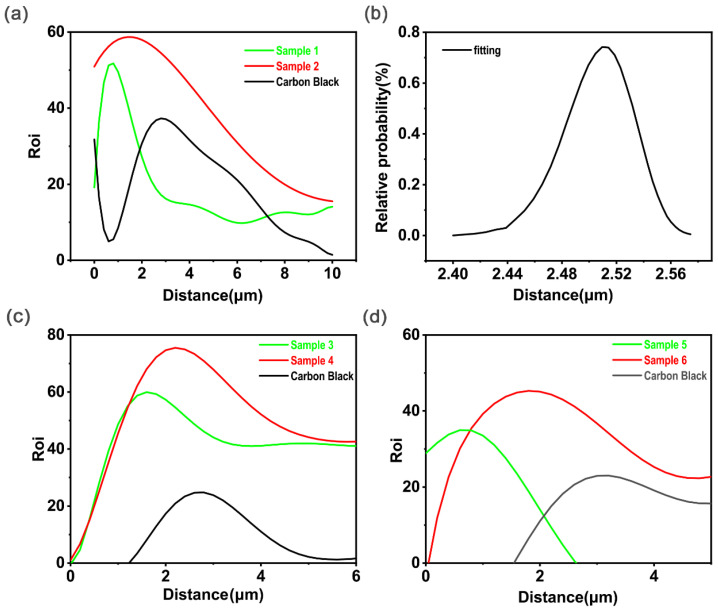
(**a**) Cross-sectional EDS images of samples 3 and 4 after ion implantation with experimental measurements. (**b**) Model-calculated distribution of carbon black particle concentration in the sample cross section after ion implantation. (**c**,**d**) Cross-sectional EDS images of samples 1, 2, 5, and 6 after ion implantation with experimental measurements.

**Table 1 polymers-15-00270-t001:** The proportions for Ag implantation energy, E, charge state, Dh, and each charge state, Or.

Dh	E/keV	Or/%
1+	40	13
2+	80	61
3+	120	25
4+	160	1

**Table 2 polymers-15-00270-t002:** The different initial velocities, the penetration distances, and the probabilities of silver ions in different events.

Event	$n_{1}$	$n_{2}\;$	$n_{3}$	Penetration Range/m	Probability/P
1	0	0	150	2.6516 × 10^−6^	4.9091 × 10^−91^
2	0	1	149	2.6515 × 10^−6^	1.8262 × 10^−88^
…	…	…	…	…	…
11,475	149	1	0	1.5519 × 10^−6^	8.8317 × 10^−131^
11,476	150	0	0	1.5309 × 10^−6^	1.2345 × 10^−133^

**Table 3 polymers-15-00270-t003:** Properties of the experimental materials.

PDMS	Elongation at break	Dielectric strength	Volume resistance
	400%	12 kV/mm	10^14^ Ω × cm
EC-600JD	Density	Specific surface area	Moisture
	0.12 g/cm^3^	1400 m^2^/g	<0.5%

## Data Availability

Data are available on request from the authors.

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
