# Peer review of "Study of a Mixed Conductive Layer Fabricated by Ion Implantation and Distribution Theory"

_polymers, 2023, doi:10.3390/polym15020270_

Round 1

Reviewer 1 Report

Dear Authors,

The manuscript presented for review entitled "Study of the mixed conductive layer fabricated by ions implantation and distribution theory" by Xuerui Fan, Huiyan Zhang, Yi Wei, Yao Huang, Huimei He, Yun Wang, Qingyun Meng and Wenjie Wu from Beijing University of Chemical Technology relate to theoretical and experimental research in the field of electrode fabrication, which can provide both stability and flexibility by using silver ions with an energy of 1640keV that have been implanted into the surface of polydimethylsiloxane (PDMS).

The work is interesting, contains all the necessary elements, has a correct general structure and the presented content is adequate to the formulated title, the research methodology is correct and relatively well described. Literature sources are appropriate to the topic of the work, although they should be supplemented with a few items from Polymers, otherwise the journals may not be appropriate for this type of work.

Authors should emphasize the novelty and originality of their research and what their results bring to the field they represent.

In addition, Authors should correct the manuscript in accordance with the following points:

1. Under Figures 1, 2, 3 and 4, a description should be added to explain what is presented in Fig. A and what is presented in Fig. B.

2. All drawings should be same, axis description should have text, symbol and unit, please improve all drawings quality as well.

3. Please make the fonts a and b smaller for the drawings.

4. Point 4.1 should not begin with a drawing but with text.

5. Point 4.2.1 should not begin with a drawing but with text.

6. Point 4.2.2 should not begin with a drawing, but with text.

7. Point 3. Start with a capital letter and use a space after the period.

8. Please explain why it was assumed that carbon black particles were rigid balls and not flakes / plates?

9. Point 3.1 concerning materials should be extended to present their properties, eg in the form of tables.

10. The content of point 3, 4 should not only refer to direct observations resulting from the obtained results but also present a discussion with the results observed by other researchers.

11. In the content of the manuscript, when providing information about materials and devices, in addition to the name, their manufacturer, city and country should be indicated - this is not always the case, it should be completed.

12. Double-check the manuscript to make sure it is in full compliance with the journal guidelines.

With these corrections made, the manuscript can be published in the journal Polymers.

Best regards,

Author Response

Reviewer #1:

The manuscript presented for review entitled "Study of the mixed conductive layer fabricated by ions implantation and distribution theory" by Xuerui Fan, Huiyan Zhang, Yi Wei, Yao Huang, Huimei He, Yun Wang, Qingyun Meng and Wenjie Wu from Beijing University of Chemical Technology relate to theoretical and experimental research in the field of electrode fabrication, which can provide both stability and flexibility by using silver ions with an energy of 1640keV that have been implanted into the surface of polydimethylsiloxane (PDMS).

The work is interesting, contains all the necessary elements, has a correct general structure and the presented content is adequate to the formulated title, the research methodology is correct and relatively well described. Literature sources are appropriate to the topic of the work, although they should be supplemented with a few items from Polymers, otherwise the journals may not be appropriate for this type of work.

Authors should emphasize the novelty and originality of their research and what their results bring to the field they represent.

In addition, Authors should correct the manuscript in accordance with the following points:

Response: Thank you for these constructive comments. And we are pleased to hear and greatly grateful to the positive overall comments on our work. Your comments are valuable and very helpful for us to revise and improve our paper.

Comments 1) Under Figures 1,2,3 and 4, a description should be added to explain what is presented in Fig A and what is presented in FigB.

Response: Thank you for this constructive comment. We had added some description of those Figures in the revised manuscript,

Comments 2) All drawings should be same, axis description should have text, symbol and unit, please improve all drawings quality as well.

Response: Thank you for these constructive comments. We have modified all drawings in the revised manuscript. 

Comments 3) Please make the fonts a and b smaller for the drawings.

Response: Thank you for this constructive comment. And we have modified all the fonts in revised manuscript.

Comments 4) Point 4.1 should not begin with a drawing but with text.

Response: Thank you for this constructive comment. And we have modified the order of discussion in revised manuscript.

Comments 5) Point 4.2.1 should not begin with a drawing but with text.

Response: Thank you for this constructive comment. And we have modified the order of discussion in revised manuscript.

Comments 6) Point 4.2.2 should not begin with a drawing but with text.

Response: Thank you for this constructive comment. And we have modified the order of discussion in revised manuscript.

Comments 7) Point 3. Start with a capital letter and use a space after the period.

Response: Thank you for this constructive comment. And we have modified the title in revised manuscript.

Comments 8) Please explain why it was assumed that carbon black particles were rigid balls and not flakes/plates?

Response: Thank you for this constructive comment. Before build model of carbon black particles, we take SEM of them. From the Figure1c which we used, we can see that black carbon particles are balls.

Fig. 1 Electron micrograph: (a) N220 carbon black, (b) 7011 carbon nanotubes, (c) 600JD highly conductive carbon black

Comments 9) Point 3.1 concerning materials should be extended to present their properties, eg in the form of tables.

Response: Thank you for this constructive comment. And we have added the table in revised manuscript.

Comments 10) The content of point 3,4 should not only refer to direct observations resulting from the obtained results but also present a discussion with the results observed by other researchers.

Response: Thank you for this constructive comment. The system studied in the manuscript was more in the field of plastic, but less in the field of rubber, so there are few references.

Comments 11) In the content of the manuscript, when providing information about materials and devices, in addition to the name, their manufacturer, city and country should be indicated-this is not always the case, it should be completed.

Response: Thank you for this constructive comment. And we have modified it in revised manuscript.

Comments 12) Double-check the manuscript to make sure it is in full compliance with the journal guidelines.

Response: We thank you for this constructive comment. And we have checked the manuscript again.

Reviewer 2 Report

The paper present the development of rubber based conductive material. These type of materials are widely used in electronic application. In order to obtain electrodes with high conductivity and stability, a metal vapour vacuum arc ion source was employed to implant silver ions on the poly-siloxane based substrate. surface,

In my opinion the document needs some major revisions as following reported:

- Figure 1, 2, 3 the authors should increase the character of main step on the axis and to use micro meters unit instead of meters. Moreover the details of difference of a, b plots should be add in the figure caption.

- In my opinion the authors should add some discussion about the electronic noise due to the electron mobility in material. This type of material could show unneglectable effects and limit of conductive at low frequency. These limitations could affect on the using of such material in electronic applications.

- The author should add some comments about the possibility of materials degradation due to the overheating phenomena during the implantation process.

- The authors should add more discussion about the material stability (fig.7) about conductivity changes and the role that plays the carbon black deposition.

Author Response

Reviewer #2:

This paper demonstrates volatile organic compound free unibody based triboelectric nanogenerator, serving as both an electric generator and a gait sensor.  The materials are common and the TENG The paper present the development of rubber based conductive material. These types of materials are widely used in electronic application. In order to obtain electrodes with high conductivity and stability, a metal vapour vacuum arc ion source was employed to implant silver ions on the poly-siloxane based substrate. surface,

In my opinion the document needs some major revisions as following reported:

Response: Thank you for your careful reading of our manuscript and your important comments, which are all valuable and very helpful for us to revise and improve our paper.

Comments 1) Figure 1,2,3 the authors should increase the character of main step on the axis and to use micro meters unit instead of meters. Moreover the details of difference of a,b plots should be add in the figure caption.

Response: Thank you for this constructive comment. And we modified it in revised manuscript.

Comments 2) In my opinion the authors should add some discussion about the electronic noise due to the electron mobility in material. This type of material could show unneglectable effects and limit of conductive at low frequency. These limitations could affect on the using of such material in electronic applications.

Response: Thank you for this constructive comment. We conducted some experiments of dialectical performance and added the results in the main text.

Comments 3) The author should add some comments about the possibility of materials degradation due to the overheating phenomena during the implantation process.

Response: Thank you for this constructive comment. During ion implantation, part of the rubber absorbs energy and turns it into gas. Therefore, only the surface topography will be changed, the performance of the material will not change.

Comments 4) The authors should add more discussion about the material stability (fig.7) about conductivity changes and the role that plays the carbon black deposition.

Response: Thank you for this constructive comment. We have added these discussions in point 4.1.1.

Round 2

Reviewer 1 Report

Dear Authors,

Thank you very much for considering the Reviewer's suggestions. Now the manuscript looks even better than before, inaccuracies have been clarified, elements have been added and developed, increasing the scientific value. The article can be printed as is.

Best regards,

Reviewer 2 Report

The authors have answered to the comments and they have added an extra experiment to discuss about my questions. In my opinion the paper is ready to be published.